# Cytokine induced inflammatory bowel disease model using organ-on-a-chip technology

**Christine Tataru**[ID]¹*, **Maya Livni**¹, **Carrie Marean-Reardon**³, **Maria Clara Franco**[ID]³,⁴, **Maude David**[ID]¹,²

**1** Oregon State University, College of Science, Microbiology, Corvallis, OR, United States of America, **2** Oregon State University, College of Pharmacy, Corvallis, OR, United States of America, **3** Oregon State University, College of Science, Biochemistry and Biophysics, Corvallis, OR, United States of America, **4** Florida International University, Herbert Wertheim College of Medicine, Center for Translational Science, Port St. Lucie, FL, United States of America

* tataruc@oregonstate.edu

**Data Availability Statement:** All relevant data are within the paper and its Supporting information files.

## Abstract

Over 2 million people in North America suffer from inflammatory bowel disease (IBD), a chronic and idiopathic inflammatory condition. While previous research has primarily focused on studying immune cells as a cause and therapeutic target for IBD, recent findings suggest that non-immune cells may also play a crucial role in mediating cytokine and chemokine signaling, and therefore IBD symptoms. In this study, we developed an organ-on-a-chip co-culture model of Caco2 epithelial and HUVEC endothelial cells and induced inflammation using pro-inflammatory cytokines TNF-α and IFN-γ. We tested different concentration ranges and delivery orientations (apical vs. basal) to develop a consistently inducible inflammatory response model. We then measured pro-inflammatory cytokines and chemokines IL-6, IL-8, and CXCL-10, as well as epithelial barrier integrity. Our results indicate that this model 1. induces IBD-like cytokine secretion in non-immune cells and 2. decreases barrier integrity, making it a feasible and reliable model to test the direct actions of potential anti-inflammatory therapeutics on epithelial and endothelial cells.

## Introduction

### Inflammatory bowel disease

The prevalence of inflammatory bowel disease (IBD) in North America ranges from 0.3% to 0.5%, and incidence is increasing starkly, especially in younger age groups [1]. IBD is divided into two main categories: ulcerative colitis (UC) and Crohn's disease (CD). The disease can be debilitating; 32.9% of patients with Crohn's disease eventually become disabled by the disease and 80% require at least one surgery in their lifetime, while between 5 and 15% of patients with UC require a colectomy to remove all or part of the colon [2, 3]. Despite the availability of multiple drug options such as amino salicylates, glucocorticoid (GC), immunosuppressive (such as azathioprine methotrexate), and TNF-α monoclonal antibodies, a large proportion of patients either do not respond or lose response to therapy [4]. Therefore, there is an urgent need to understand the molecular mechanisms underlying the pathogenesis of IBD to develop new therapeutic approaches to treat this debilitating disease.

**Funding:** Oregon State University College of Science SCiRisIII funding: https://internal.science.oregonstate.edu/rdu/internal-research-funding-program, Oregon State University Microbiology Department New Professor Start-up funding, Oregon State University Research Equipment Reserve Fund, Larry W. Martin & Joyce B. O'Neill Endowed Fellowship The funders had no role in study design, data collection and analysis, decision to publish, or preparation of the manuscript.

## Non-immune cell inflammatory responses present an opportunity for new and localized therapeutics

Since 1997, researchers have recognized that "epithelial cells play a critical role in the inflammatory response" [5]. Epithelial cells manufacture and secrete cytokines and chemokines to regulate inflammation and signal recruitment of immune cells, and they synthesize growth factors with autocrine actions to facilitate their own repair [5]. It has been suggested that epithelial cells are an important source of pro-inflammatory cytokine signaling and may be used to control intestinal homeostasis to some extent [6]. Their influence over inflammatory signaling makes them particularly interesting targets for alleviating symptoms of IBD, and pausing the iconic feed-forward cycle of inflammation characteristic of the disease.

Intestinal epithelial cells (IECs) change their inflammation-related expression profiles in the context of IBD. For example, Toll-like receptors (TLR), critical innate immune system pathway regulators, are known to exhibit altered expression profiles in IBD patients. Specifically, TLR4 is overexpressed, and receptors shift from the basolateral side of cells to the apical side. In contrast, TLR2 and TLR5 are expressed at significantly lower levels in IBD [6]. Colonic epithelial cells isolated from active IBD patients secrete the neutrophil-attracting cytokine IL-8, which then exacerbates the harmful feed-forward cycle of inflammation observed [7, 8]. IL-21R has also been observed to be upregulated in IECs of IBD patients; this cytokine leads to increased CCL20 synthesis, which subsequently attracts T cells and dendritic cells [9, 10]. The IL-22 inflammatory cascade has also been implicated in IBD, wherein IL-22 is increased, leading to the dysregulation of intestinal epithelial cell (IEC) proliferation and migration, and stimulating IEC secretion of IL-1, TNF-$\alpha$, IL-6, and IL-8 [11].

Epithelial cells may also play a major role in adaptive immune response. These cells express the antigen-presenting major histocompatibility complex II (MHCII) mostly on the villi, but in IBD, MHCII is instead upregulated in the small intestinal crypts, a change mostly driven by IFN-$\gamma$ [12, 13]. *In vitro* studies have shown that suppressor T cells can be signalled to proliferate by antigen-presenting epithelial cells, contributing to the development of oral tolerance [14].

It is also clear that T cell originating cytokines IFN-$\gamma$ and TNF-$\alpha$ can control epithelial cell properties by multiple mechanisms including disrupting barrier function (i.e. increasing gut permeability) [15, 16]. Disruption of barrier function is one of the hallmarks of IBD; poor barrier function can lead to antigen leakage into the lamina propria, creating inflammatory responses of massive proportions. While the direct cause of IBD remains unknown, growing evidence suggests it may be caused by a disruption in the finely controlled equilibrium between host response to antigens in the intestinal epithelium [17]. Therefore, there is an urgent need for the development of models that reflect this disruption in equilibrium.

## *In vitro* models of non-immune cell inflammatory response

*In vitro* models enable the investigation of the precise function of candidate molecules to unravel cellular mechanisms and identify novel specific targets [5]. Existing *in vitro* models have been indispensable in developing the current IBD therapies, however, they remain insufficient as they largely focus on very specific, rather than general, abnormalities and mechanisms [17].

The most common *in vitro* models of IBD involve treating immortalized cell lines (e.g. Caco2, HT29, T84) with high concentrations of dextran sodium sulfate (DSS) or lipopolysaccharide (LPS) [17]. Though often utilized *in vivo*, both DSS and LPS can be used *in vitro* to elicit many of the same phenotypes; treatment with DSS is directly toxic to colonic epithelium; it causes ulcers, loss of crypts, and infiltration of granulocytes, all of which are characteristic of

IBD [18–20]. This model lends itself to the study of factors that maintain or reestablish barrier integrity, or factors that protect against toxicity. However, it should not be considered an accurate mimic of human IBD, as it assumes a specific mechanism of origin, massive epithelial damage, which is unlikely to be the cause of disease in patients [18]. LPS treatment, or alternatively infection with invasive *E. coli* or *Salmonella*, activates extreme innate immune responses to bacterial components as a way of simulating IBD. This model lends itself to the study of factors that prevent or inhibit infection, but should not be considered an accurate model as it assumes an infectious disease origin of IBD [18, 21, 22].

Cytokine-based, non-immune cell models of IBD do also exist, though they mainly focus on barrier integrity measurements in epithelial cells. Amasheh et. al treated primary rat colon cells with TNF-$\alpha$ and IFN-$\gamma$ in Ussing chambers, and showed that changes in barrier function and mucosal structure did occur, as would be expected in IBD [23]. However, the use of cells from a different species limits the generalizability of these conclusions to humans. Others have shown that treating Caco-2 cells co-cultured with macrophages with IL-1$\beta$ and IFN$\gamma$ results in a decrease in transepithelial electrical resistance (TEER), which indicates a decrease in barrier integrity [24]. Dosh et. al report that Caco-2 and HT29-MTX co-cultures grown on hydrogel scaffolds and treated with IL-1$\beta$ and TNF-$\alpha$ exhibit changes in tight junction protein expression (ALP and ZO-1), as well as an increase in apoptosis signaling (caspase 3), both of which may indicate a decrease in barrier function [25]. As discussed above, non-immune cells play a vital role in cytokine and chemokine signaling that is not limited to barrier function.

Despite the role epithelial cells play in the inflammatory response, there is currently a dearth of *in vitro* models for the study of factors that interrupt or reduce the non-immune cytokine response characteristic of IBD. Here, we present a gut-on-a-chip co-culture model that stimulates non-immune cells with T-cell originating pro-inflammatory cytokines to simulate the non-immune cytokine response.

## Gut on a chip

Organ-on-a-chip models are robust platforms to explore pathophysiological mechanisms in multiple diseases. In this model category, multiple cell types are co-cultured and subjected to tissue microenvironmental forces such as dynamic flow and/or mechanoactuation [26]. In the case of the gut on a chip, the result is a more physiologically relevant system that recapitulates gut architecture (crypts and villi-like structures), gene expression, and infection dynamics [26–29]. Lastly, cells in this system can be cultured for up to 21 days, enabling the study of longer-term therapeutic effects [26].

## Results

In this study, Caco-2 cells and HUVEC cells were co-cultured on opposite sides of a microporous membrane in a gut-on-a-chip system. Cells were subjected to a constant flow of cell-specific media, along with periodic stretch forces on the microporous membrane mimicking peristalsis. Pro-inflammatory cytokines TNF-$\alpha$ and IFN-$\gamma$ were added in concentrations of 25 or 100 $\frac{ng}{ml}$ to the top (Caco2-cultured) or bottom (HUVEC-cultured) channel, and the responding secretion of IBD-relevant cytokine and chemokines IL-6, IL-8, and CXCL-10 was measured using an ELISA assay. Cascade blue fluorescent marker was used to measured barrier permeability during inflammatory challenge (Fig 1). This experimental system enables the interrogation of molecules or microbial species with potential anti-inflammatory properties.

We found that 25 $\frac{ng}{ml}$ of each cytokine in the bottom channel (with HUVEC cells) resulted in a significant and consistent inflammatory response from all three cytokines/chemokines

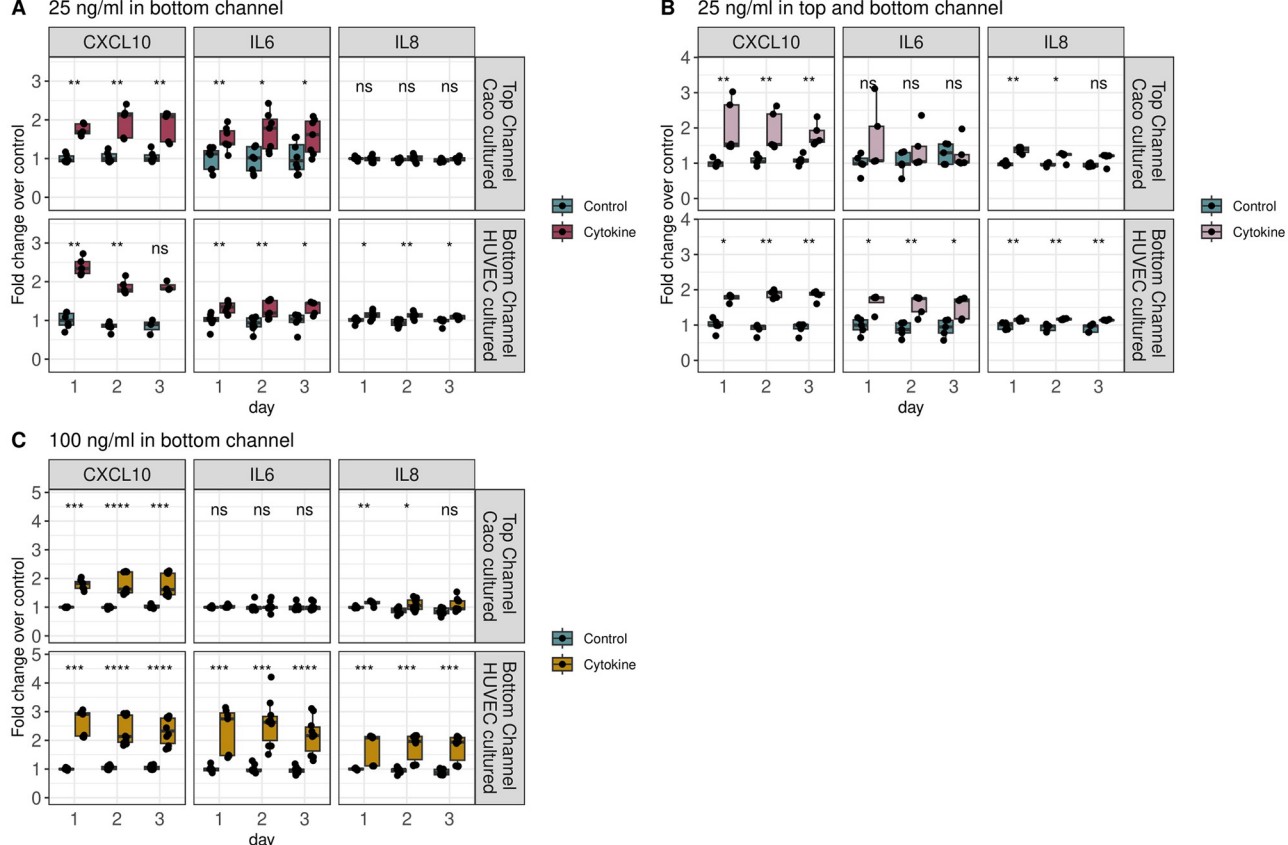

**Fig 1. Experimental setup.** Caco2 and HUVEC cells were grown on opposite sides of a microporous membrane. Cells were treated with different concentrations of pro-inflammatory cytokines in either the top or bottom channel, and a cascade blue fluorescent permeability tracer was added to the top channel to quantify cell layer permeability. The levels of CXCL-10, IL-6, IL-8 and permeability tracer in outlet media were then measured.

(CXCL-10, IL-6, and IL-8) in the bottom channel, as well as an increase in CXCL-10 and IL-6 (but not IL-8) in the top channel (Fig 2A, Table 1). This response was most evident during the first two days after treatment, and in the case of CXCL-10 in the bottom channel, became statistically insignificant by the third day after treatment (Fig 2A).

**Fig 2. Caco and HUVEC cytokine response to inflammatory challenge.** CXCL-10, IL-6, and IL-8 secretion in top/bottom channels after treatment with: (A) 25 $\frac{ng}{ml}$ of TNF-$\alpha$ and IFN-$\gamma$ in the bottom channel (B) 25 $\frac{ng}{ml}$ of TNF-$\alpha$ and IFN-$\gamma$ in the top and bottom channel (C) 100 $\frac{ng}{ml}$ of TNF-$\alpha$ and IFN-$\gamma$ in the bottom channel. Cells were treated with pro-inflammatory cytokines on Day0, and concentrations are reported as log fold change over untreated controls. P-values determined using Wilcox rank sum test vs. control ($p < .05$ = *, $p < 0.01$ = **, $p < 0.001$ = ***, $p < 0.0001$ = ****).

**Table 1. Caco and HUVEC cell cytokine response to inflammatory challenge.** Values are reported as fold change over median control values. Averages and standard deviations were calculated across all 3 days measured per condition. Significance values per day as shown in Fig 2.

| Channel | Cytokine | $25 \frac{ng}{ml}$ bottom | | $25\frac{ng}{ml}$ top and bottom | | $100 \frac{ng}{ml}$ bottom | |
|---|---|---|---|---|---|---|---|
| | | Avg. | S.D. | Avg. | S.D. | Avg. | S.D. |
| Top channel | CXCL10 | 1.846 | 0.327 | 1.915 | 0.536 | 1.792 | 0.325 |
| Top channel | IL6 | 1.516 | 0.380 | 1.436 | 0.643 | 1.020 | 0.119 |
| Top channel | IL8 | 0.964 | 0.060 | 1.234 | 0.161 | 1.098 | 0.181 |
| Bottom channel | CXCL10 | 2.070 | 0.320 | 1.833 | 0.121 | 2.408 | 0.471 |
| Bottom channel | IL6 | 1.362 | 0.195 | 1.566 | 0.276 | 2.362 | 0.739 |
| Bottom channel | IL8 | 1.164 | 0.061 | 1.153 | 0.027 | 1.746 | 0.464 |

$25 \frac{ng}{ml}$ of each cytokine in the top and bottom channel resulted in a similar phenotype in the bottom channel, with a significant increase in all three cytokines elevated throughout all three days (Fig 2B). In the top channel, we also observed a significant increase in CXCL-10 and IL-8, but not IL-6 (Fig 2B, Table 1). In this case, the response of all secondary cytokines was also most evident directly after treatment with pro-inflammatory stimulus and decreased in the days following treatment (Fig 2B).

Upon increasing the concentration of pro-inflammatory cytokines to $100 \frac{ng}{ml}$ in the bottom channel, we observed the expected stark increase in all three response cytokine/chemokines in the bottom channel. In the top channel, CXCL-10 and IL-8 levels, but not IL-6 levels, significantly increased compared to controls. (Fig 2C, Table 1).

In comparing relative effects of treatments, we found that increasing concentration of stimulating cytokine in the bottom channel (25 vs. $100 \frac{ng}{ml}$) increased the secretion of CXCL-10, IL-6, and IL-8 in the bottom channel, increased the secretion of IL-8 in the top channel, but decreased the secretion of IL-6 in the top channel (S1–S3 Figs). Increasing the concentration of stimulating cytokines in the top channel ($25 \frac{ng}{ml}$ bottom vs $25 \frac{ng}{ml}$ top and bottom) had no consistent significant effect on the bottom channel cytokines, and increased IL-8 secretion in the top channel (S1–S3 Figs).

To compare these results with those obtained from a traditional cell culture model, we independently incubated HUVEC or Caco2 cells in the presence or absence of varying concentrations of TNF-$\alpha$ and IFN-$\gamma$ in a 96-well plate. For both cell types, $25 \frac{ng}{ml}$ of each stimulus cytokine was sufficient to produce near maximum response in every cytokine/chemokine, except for IL-6 in Caco2 cells, which required $75 \frac{ng}{ml}$ to produce maximum response (S4 Fig). Additionally, no concentration of cytokine resulted in cell significant cell death as measured by a crystal violet assay (S5 Fig).

## Permeability

We then tested the apparent permeability of cell layers using the Cascade Blue fluorescent marker as a permeability tracer. Cascade Blue cannot be metabolized, allowing us to test the permeability of the cell layer by looking at the ratio of salt in the top vs. bottom channels. The marker was loaded into inlet reservoirs on Day 1, and quantity of marker found in the bottom channel outlet media at the indicated times was used to assess barrier function (see Methods). We found that $25 \frac{ng}{ml}$ added to either the bottom or the top and bottom channel did not result in significant changes in barrier function. However, $100 \frac{ng}{ml}$ of pro-inflammatory cytokines added to the bottom channel did increase apparent permeability significantly and consistently across all 3 days measured (Fig 3).

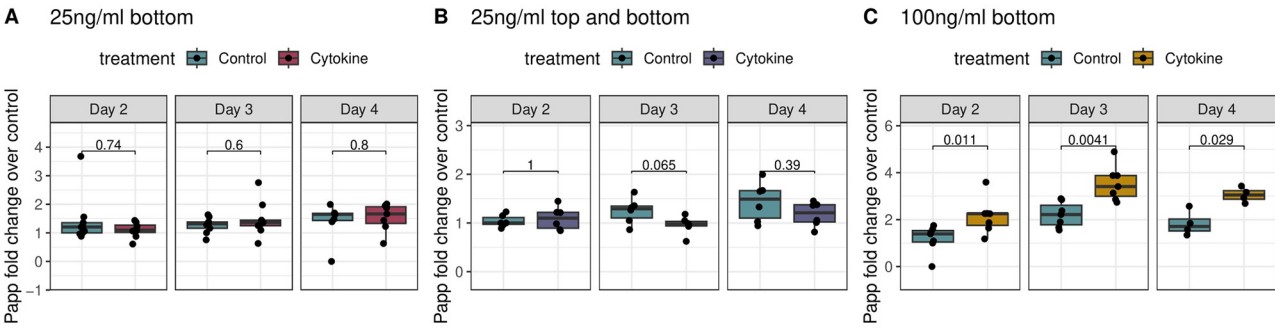

**Fig 3. Caco and HUVEC change in permeability due to inflammatory challenge.** Apparent permeability after treatment with: (A) 25 $\frac{ng}{ml}$ of TNF-$\alpha$ and IFN-$\gamma$ in the bottom channel (B) 25 $\frac{ng}{ml}$ of TNF-$\alpha$ and IFN-$\gamma$ in the top and bottom channel (C) 100 $\frac{ng}{ml}$ of TNF-$\alpha$ and IFN-$\gamma$ in the bottom channel. Cells were treated with pro-inflammatory cytokines on Day1, and concentrations are reported as log fold change over untreated controls.

## Discussion

### Model usage

The described model was developed using the Emulate organ-on-a-chip basic research kit (Emulate, MA).We report here the cytokine secretion and permeability characteristics of Caco2 epithelial and HUVEC endothelial cells treated with TNF-$\alpha$ and IFN-$\gamma$ in different concentrations and in different orientations on a gut-on-a-chip. Caco2 cells were seeded in the top channel, and HUVEC cells in the bottom channel. As such, responsive cytokines/chemokines found in the bottom channel may be produced by HUVEC cells, and/or by basolateral secretion from Caco2 cells. Responsive cytokines/chemokines found in the top channel may be produced by apical secretion from Caco2 cells, or by diffusion of HUVEC-produced cytokines through a disrupted Caco2 cell barrier. Overall, we found that 25 $\frac{ng}{ml}$ of cytokines in the bottom channel produced CXCL-10 and IL-6 in both channels, but IL-8 only in the bottom channel; 25 $\frac{ng}{ml}$ added to both channels produced CXCL-10 and IL-8 in both channels, but IL-6 only in the bottom channel; and 100 $\frac{ng}{ml}$ in the bottom channel triggered production of CXCL-10 and IL-8 in both channels, but IL-6 only in the top channel.

100 $\frac{ng}{ml}$ in the bottom channel was the only treatment that also disrupted barrier function as is seen in inflammatory bowel disease. Further investigations are required to determine the molecular mechanisms of this observed loss in barrier function, those it was determined that increased permeability was not cell death-related.

In conclusion, we recommend the use of this system with 25 $\frac{ng}{ml}$ in the top and bottom channel for investigation of factors surrounding cytokine/chemokine production, as this treatment resulted in the most consistent secondary cytokine response in both channels. We recommend the use of 100 $\frac{ng}{ml}$ in the bottom channel for the investigation of factors surrounding barrier integrity of gastrointestinal cells, as this treatment was the only one observed to disrupt barrier function.

### Spatial considerations

It is worthwhile to note the spatial considerations of the model presented. 25 $\frac{ng}{ml}$ of cytokine stimulus in the bottom channel produced IL-8 in the bottom but not the top channel. This is consistent with previous literature, which found that Caco-2 cells exhibit polarized IL-8 secretion, secreting apical IL-8 when treated apically, and both basal and apical IL-8 when treated basolaterally [30]. Given higher TNF$\alpha$ concentrations found in the intestinal lumen in IBD, it

is relevant to explicitly differentiate between the luminal and circulating concentrations of IL-8 [31]. Additionally, contrary to expectation, IL-6 was not increased in the top channel when high concentrations of cytokine are added to the bottom channel ($100 \frac{ng}{ml}$), but was observed when only $25 \frac{ng}{ml}$ was added to the bottom channel. This may be related to the dual role of IL-6 as a pro- and anti-inflammatory cytokine; in chronic inflammation, IL-6 is considered pro-inflammatory, while in acute inflammation, IL-6 is known to be protective against aberrant inflammation [32–34]. We hypothesize that epithelial cell-secreted IL-6, in comparison to that produced by macrophages and other immune cells, may only be produced and secreted apically under low pro-inflammatory stimulus. This is consistent with work by Cudicini et al. which finds that Sertoli cells, as subtype of epithelial cell, are found to predominantly secrete IL-6 apically [35]. CXCL-10 exhibited no differences in terms of vectorial secretion, and was also the most consistently secreted under all conditions. Further investigation culturing single cell-types within the dynamic chip environment (e.g. mechanical and flow forces) is necessary to fully characterize the vectorial secretion of cytokines as well to determine the effect of cell-cell interactions on inflammatory response.

## Cytokines relevant to this model

In this work, we report on a model investigating the non-immune cell inflammatory response as it relates to IBD. We induced inflammation in epithelial and endothelial cells using major pro-inflammatory cytokines observed in IBD, such as TNF-$\alpha$ and IFN-$\gamma$, then measured the secretion of other cytokine/chemokines relevant to IBD, including CXCL-10, IL-6, and IL-18.

Elevated IFN-$\gamma$ levels are characteristic in CD, and prevention of IFN-$\gamma$ production can significantly alleviate disease symptoms [15].

TNF-$\alpha$ has a significant function in IBD pathogenesis by increasing IL-1$\beta$, IL-6, and IL-33 expression. The clinical severity of UC and CD correlates with TNF-$\alpha$ levels in the serum of IBD patients [36, 37]. Anti-TNF therapy is very effective at treating IBD symptoms, leading to mucosal healing, reduced hospitalizations and surgeries, and improved patient quality of life [38]. Anti-TNF agents are more effective in patients with a shorter disease history, and are often used earlier during IBD therapy [39].

On the other hand, IL-8 is a ubiquitous neutrophil attracting chemokine that has been proposed as a potential diagnostic or prognostic marker for several GI disease states [40]. It is well established that IL-8 production is increased in the tissue of IBD patients, especially UC patients, compared with healthy controls [41–44]. As a result, therapeutics that target IL-8 are currently under development. One such example is an IL-8 blocking antibody shown to significantly inhibit the recruitment of neutrophils and decrease markers of inflammation in patients with palmoplantar pustulosis (PPP), a rare chronic inflammation of the skin. This finding suggests the potential use of anti-IL-8 therapeutics in treating IBD [45].

IL-6 is a main proinflammatory cytokine that activates signal transducer and activator of transcription 3 (STAT3), and has a key role in the pathogenesis of UC and the carcinogenesis of colorectal cancers related to UC [46]. STAT3 expression is induced by TNF-$\alpha$, and STAT3 levels are increased in the tissues of ulcerative colitis and Crohn's disease patients [47–49]. Blocking IL-6 signal transduction led to clinical remission in 19% of patients in a small clinical trial, suggesting that decreasing IL-6 response may be a promising therapeutic avenue [50]

CXCL-10 is a chemokine characteristic of IBD that attracts Th1-polarized effector T cells to inflamed sites. There is a positive feedback loop in which IFN-$\gamma$ induces the production of this chemokine, which then recruits T cells that produce IFN-$\gamma$ [51]. CXCL-10 levels are increased in active IBD, both UC and CD, mainly localized to mucosal epithelial cells, and its expression is inducible by IFN-$\gamma$ [52]. Treatment with anti-CXCL10 during colitis in mice decreased

clinical and histological disease severity and lowered mononuclear and TH1 cell recruitment, suggesting its promise as a therapeutic target for IBD [53].

## Advantages and limitations of the described system

The co-culture system presented here provides a framework through which small molecules, microbial products, or even live biologic therapeutics may be tested in vitro to investigate inflammatory action and mechanism of action in epithelial and endothelial cells. Different phenotypes can be created by simply changing the concentration, orientation, and type of inflammatory cytokine, and multiple output metrics (e.g. cytokine concentration, permeability characteristics) may be measured over multiple days. The system is simple, highly reproducible, and does not assume any original cause of inflammation. This flexibility enables the study of mechanisms relevant to multiple IBD-subtypes, and may be expanded to other applications, such as inhibiting cytokine storm phenomenon in other diseases like SarsCov2, where non-immune cell cytokine secretion play a critical role [54, 55]. We showed here the use of this model to report cytokine secretion of epithelial and endothelial cells as markers of response to inflammatory stimuli. However, the system could be adapted for the assessment of other relevant responses, such as changes in MHCII expression patterns, differentiation state of the epithelium, TLR expression, and gap junction protein expression. Further characterization of the model is necessary to develop a full platform on which to screen anti-inflammatory therapeutics or microbial products. Importantly, this gut-chip model reconstitutes elements of IBD, and presents conditions that better resemble the bowel as a dynamic barrier subject to mechanical forces in comparison to traditional 2D systems.

**Advantages of co-culture.** It has been previously shown that epithelial cells co-cultured with endothelial cells on opposite sides of a membrane present different physiological responses than those cultured alone. For instance, Jin et al. found that Caco2 cell co-cultured with HUVEC cells express significantly more aminopeptidase catalytic activity and villin, both markers for epithelial differentiation. They also observed significantly increased coverage of glycocalyx, a dense gel-like meshwork that surround the cell and acts as a physical protective barrier [56]. In another study using human biopsy derived colonoids (instead of Caco2) and Colonic Human Intestinal Microvascular Endothelial Cells (cHIMECs) (instead of HUVECs), Apostolou et al. showed that the inclusion of endothelial cells significantly decreases permeability of the colonic cell layer, and upregulates the expression of genes related to morphogenesis of colonic epithelial barrier such as cytoskeletal organization. They also found that transcriptomic profiles from colonic cells co-cultured with endothelial cells clearly cluster together as compared to profiles of cells cultured alone [57], highlighting the relevance of using co-cultured models to better recreate the pathophysiological characteristic of the disease process.

This gut on a chip system presents the ability to co-culture not just epithelial and endothelial cells, but a battery of other cell types such as gut bacteria as well as immune cells [58, 59]. This work represents a foundational framework upon which more complex networks of interactive cell types may be built upon.

## Conclusion

In conclusion, this study established a gut inflammation co-culture in vitro model using Caco-2 epithelial and HUVEC endothelial cells stimulated with pro-inflammatory cytokines TNF-$\alpha$ and IFN-$\gamma$ for the study of IBD pathology. The co-culture model replicated multiple molecular hallmarks of IBD, including non-immune cell cytokine and chemokine secretion (CXCL-10, IL-6, and IL-8) as well as changes in epithelial barrier integrity. We argue that induction of

**Table 2. Key resource table.**

| Component | Company | Catalog number |
|---|---|---|
| Caco2 cells | ATCC | CRL-2102 |
| HUVEC cells | Lonza | C2519A |
| DMEM | ThermoFisher Scientific | 10566024 |
| Penicillin-Streptomycin | Sigma-Aldrich | P0781 |
| EGM2 Endothelial cell growth media bullet kit | Lonza | CC-3162 |
| Matrigel | VWR | 47743–715 |
| Type1 Collagen | VWR | 47747–218 |
| FBS | Sigma-Aldrich | F4135 |
| Trypsin-EDTA | Sigma-Aldrich | T3924 |
| Cascade Blue fluorescent salt | ThermoFisher Scientific | C687 |
| IL-8 ELISA kit | R&D Systems | DY208 |
| IL-6 ELISA kit | R&D Systems | DY206 |
| CXCL-10 ELISA kit | R&D Systems | DY266 |

inflammation using only pro-inflammatory cytokines TNF-$\alpha$ and IFN-$\gamma$ can create a simple and more accurate model of the cytokine secretion and barrier integrity aspects of IBD, and is less likely to induce unknown or aberrant side-effects, better reflecting the expected response to direct tissue damage or infection. We offer concentrations and techniques to replicate this inflammatory model, and suggest its use to further the development of anti-inflammatory therapies, and for the study of molecular mechanism underlying IBD pathology.

## Materials and methods

List of all reagents and sources can be found in Table 2.

### Cell lines and culture conditions

Human intestinal epithelial cell line Caco-2 cells were cultured in DMEM (high glucose, Gluta-MAX) supplemented with 20% FBS and 1% penicillin-streptomycin. Human umbilical vein endothelial (HUVEC) cells were cultured in endothelial growth media (EGM2 bullet kit). Caco2 cells were used on passage 58, HUVEC cells were used on passage 6. Cells were grown in 10 cm plates to the appropriate density for the cell type being tested, 80–100%. Cell cultures were incubated in a humidified 5% $CO_2$ incubator at 37 degrees C.

### Co-cultures of Caco-2 and HUVEC on gut-on-a-chip

Protocols provided by Emulate Inc were followed to co-culture Caco2 and HUVEC cells in the organ chip model. In greater detail:

Day -1:

To prepare chip for cultures, the microporous membrane was first activated using UV light (10 min, wash, then 5 min). The membrane was then coated with extracellular matrix (Matrigel and TypeI collagen), and allowed to incubate at 37C, 5% CO2 overnight.

Day 0:

HUVEC cells were incubated with trypsin-EDTA for 3 min. Detached cells were collected and centrifuged for 4 min at 400 g, and resuspended in fresh media to a concentration of $7x10^6$ cells / ml. Cells were then seeded onto the bottom channel of the chip, and chips were

inverted and incubated at 37 degrees C for 2 hours to allow cells to attach to the bottom side of the microporous membrane.

Caco2 cells were incubated with trypsin-EDTA for 3 min. Detached cells were collected and centrifuged for 4 min at 400 g, and resuspended in fresh media to a concentration of $3x10^6$ cells / ml. Cells were then seeded onto the top channel of the chip, and chips were again incubated at 37 degrees C, 5% CO2 for 2 hours to allow cells to attach to the top side of the microporous membrane.

After adherence, a gravity wash, wherein cells were gently washed with fresh media, was performed to give cells sufficient access to media, and chips were incubated at 37 degrees C, 5% CO2 overnight.

Day 1:

The following day, Caco2 media (DMEM + 20% FBS) and HUVEC media (EGM-2) began a continuous flow through the top and bottom channel respectively at a rate of 30 $\mu$l/hr. This flow was continued for the remainder of the experiment.

Day 2:

Peristaltic-like force stretching the central membrane 2% of its width at a frequency of 0.15 Hz was applied. Membrane was stretched at this frequency continuously from this point onward.

Day 3:

Excess media was aspirated from the reservoirs and replaced with fresh media. The concentration of FBS in HUVEC media was changed to 0.5% FBS to discourage overgrowth. Peristaltic-like force was increased to deform the membrane 10% of its width.

Day 6:

Media in the top channel was aspirated and replaced with Caco2 media supplemented 10% FBS, to discourage overgrowth in long-term co-culture.

Day 7:

Media was aspirated and replaced. In the bottom channel, HUVEC media with 0.5% FBS was mixed with appropriate cytokine for the testing condition. In the top channel, Caco2 media with 10% FBS was mixed with appropriate cytokine treatment for the testing conditions and 100 $\frac{ug}{ml}$ of Cascade Blue fluorescent permeability tracer.

Day 8–10 (Day 2–4 in the results plots):

Media from inlet and outlet ports was collected and stored at -80C. Cytokine and fluorescent marker concentrations were later measured using appropriate assays.

## Cytokine treatment

Three cytokine treatment conditions were tested: 25 $\frac{ng}{ml}$ TNF-$\alpha$ and 25 $\frac{ng}{ml}$ IFN-$\gamma$ in the bottom channel, 25 $\frac{ng}{ml}$ TNF-$\alpha$ and 25 $\frac{ng}{ml}$ IFN-$\gamma$ in both channels, and 100 $\frac{ng}{ml}$ TNF-$\alpha$ and 100 $\frac{ng}{ml}$ IFN-$\gamma$ in the bottom channel. On Day 6, Caco2 and HUVEC media were prepared as normal, top channel cytokines were added to Caco2 media, and bottom channel cytokines were added to HUVEC media. Cell culture media was placed into appropriate inlet reservoirs. Effluent media was collected on days 7, 8, and 9 (days 1–3 after cytokine treatment, as expressed in Figs 2 and 3) to measure cytokine concentration and permeability.

## Quantifying cytokine secretion

Effluent media was collected from each channel, and cytokines IL-8, IL-6, and chemokine CXCL10 were measured using an ELISA assay per manufacturer directions (R&D Systems DuoSet).

## Permeability measurements

On Day 6, Cascade Blue fluorescent salt ($100 \frac{ug}{ml}$) was added to Caco2 media (top channel). Effluent media was collected and Cascade Blue was quantified using a plate reader (excitation: 380 nm; emission: 460 nm). The following equation was used to calculate apparent permeability: $P_{app} = -\frac{Q_R * Q_D}{SA * (Q_R + Q_D)} * ln\left[1 - \frac{C_{R,0} * (Q_R + Q_D)}{Q_R * C_{R,0} + Q_D * C_{D,0}}\right]$ where $P_{app}$ with the apparent permeability in $\frac{cm}{s}$; SA is the surface area of sections of the channel that overlap ($0.17\ cm^2$); $Q_R$ and $Q_D$ are the fluid flow rates in the dosing and receiving channels ($cm^3/s$); and $C_{R,0}$ and $C_{D,0}$ are the recovered concentrations in the dosing and receiving channels respectively, in any consistent units (units will cancel out). (https://emulatebio.com/wp-content/uploads/2021/06/EP187_v1.0_Barrier_Function_Analysis_Protocol.pdf).

## Monocultures in 2D

Cells were grown in 10 cm dishes to 80–100% density. Media was removed and plates were rinsed with 10 ml DPBS or HBSS depending on the cell type. Cells were then incubated with 2 ml 0.05% trypsin for 3–5 minutes, and trypsin neutralized with 8 ml of DMEM/10% FBS. Cells were centrifuge at 400xg for 4 min and the pellet resuspended in 2 ml of the appropriate cell culture media. Cells were seeded in 96-well plates at a density of 40,000 cells/well for high density plates. Cells were initially plated in 100 $\mu l$ media/well and incubated at 37 degrees C, 5% CO2 for 2–3 hours to allow attachment before the addition of treatments. Treatments were prepared in complete media via serial dilution at a concentration of 2x the final concentration and then added to the plates at a volume of 100 $\mu l$/well. Plates were then incubated at 37C, 5% CO2 for 48 hours. Media was extracted from each well individually and cytokine concentration was measured by ELISA assay as described above.

## Supporting information

**S1 Fig. Caco2 and HUVEC CXCL-10 response by day and treatment.** Increasing concentration of stimulating cytokine in bottom channel increased response in HUVECs but not Caco2s (red vs. yellow). Adding stimulating cytokine in top channel had no measurable effect on either cell type (red vs. purple). P-values determined using Wilcox rank sum test ($p < .05 =$ *, $p < 0.01 =$ **, $p < 0.001 =$ ***, $p < 0.0001 =$ ****).
(TIF)

**S2 Fig. Caco2 and HUVEC IL-6 response by day.** Increasing concentration of stimulating cytokine in bottom channel increased response in HUVECs but not Caco2s (red vs. yellow). Adding stimulating cytokine in top channel had no measurable effect on either cell type (red vs. purple). P-values determined using Wilcox rank sum test ($p < .05 =$ *, $p < 0.01 =$ **, $p < 0.001 =$ ***, $p < 0.0001 =$ ****).
(TIF)

**S3 Fig. Caco2 and HUVEC IL-8 response by day.** Increasing concentration of stimulating cytokine in bottom channel increased response in HUVECs but not Caco2s (red vs. yellow). Adding stimulating cytokine in top channel increased response in Caco2 cells but not HUVECs (red vs. purple). P-values determined using Wilcox rank sum test ($p < .05 =$ *, $p < 0.01 =$ **, $p < 0.001 =$ ***, $p < 0.0001 =$ ****.
(TIF)

**S4 Fig. Caco2 and HUVEC cytokine response curve in traditional 2D culture.** Maximum response is seen at concentrations of stimulating cytokine 25 $\frac{ng}{ml}$ and above, with the exception

of IL-6 in Caco2 cells.
(TIF)

**S5 Fig. Caco2 and HUVEC survival curve in traditional 2D culture.** Neither cell type exhibits cell death compared to untreated cells. At some concentrations, cytokine treatment in fact increased cell growth and survival. P-values determined using Wilcox rank sum test over controls ($p < .05 = *$, $p < 0.01 = **$, $p < 0.001 = ***$, $p < 0.0001 = ****$).
(TIF)

## Acknowledgments

Thank you to Grasiella Andriani for training on the Emulate device and continued consultation. Thank you to Kyle Nguyen for training in general techniques and continued consultation.

## Author Contributions

**Conceptualization:** Christine Tataru.

**Formal analysis:** Christine Tataru.

**Funding acquisition:** Maude David.

**Investigation:** Christine Tataru, Maya Livni, Carrie Marean-Reardon.

**Methodology:** Christine Tataru, Maya Livni, Carrie Marean-Reardon, Maria Clara Franco.

**Project administration:** Christine Tataru, Maude David.

**Resources:** Maude David.

**Supervision:** Maude David.

**Validation:** Christine Tataru, Maude David.

**Visualization:** Christine Tataru.

**Writing – original draft:** Christine Tataru.

**Writing – review & editing:** Christine Tataru, Carrie Marean-Reardon.

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
