## [Decision Letter · Decision Letter 0]

20 Mar 2023

PONE-D-23-02456Cytokine induced inflammatory bowel disease model using organ-on-a-chip technologyPLOS ONE

Dear Dr. Tataru,

Thank you for submitting your manuscript to PLOS ONE. After careful consideration, we feel that it has merit but does not fully meet PLOS ONE’s publication criteria as it currently stands. Therefore, we invite you to submit a revised version of the manuscript that addresses the points raised during the review process.

We look forward to receiving your revised manuscript.

Kind regards,

Masanori A. Murayama

Academic Editor

PLOS ONE

Journal Requirements:

2. Please ensure that you have specified (1) whether consent was informed and (2) what type you obtained (for instance, written or verbal, and if verbal, how it was documented and witnessed). If your study included minors, state whether you obtained consent from parents or guardians. If the need for consent was waived by the ethics committee, please include this information.

"Oregon State University College of Science SCiRisIII funding:

https://internal.science.oregonstate.edu/rdu/internal-research-funding-program, Oregon

State University Microbiology Department New Professor Start-up funding, Oregon

State University Research Equipment Reserve Fund, Larry W. Martin & Joyce B.

O'Neill Endowed Fellowship"

"I have read the journal's policy and the authors of this manuscript have the following competing interests: MMD has financial interests in NeuroBiome LLC, Microbiome Engineering LLC and  Second genome Inc. NeuroBiome LLC develops technology related to this research. The conduct, outcomes, or reporting of this research could benefit NeuroBiome LLC and could potentially benefit MMD."

Additional Editor Comments:

Thank you for submitting your study to PLOS ONE. Reviewers have some critical comments. So I will require more adequate discussion in revised manuscript. And authors need submit an appropriate format.

Reviewers' comments:

Reviewer's Responses to Questions

**Comments to the Author**

1. Is the manuscript technically sound, and do the data support the conclusions?

Reviewer #1: Yes

Reviewer #2: No

2. Has the statistical analysis been performed appropriately and rigorously? 

Reviewer #1: Yes

Reviewer #2: Yes

3. Have the authors made all data underlying the findings in their manuscript fully available?

Reviewer #1: Yes

Reviewer #2: Yes

4. Is the manuscript presented in an intelligible fashion and written in standard English?

Reviewer #1: Yes

Reviewer #2: Yes

5. Review Comments to the Author

Reviewer #1: In this study, Tataru et al designed in vitro co-culture models of separated (by microporous membrane) epithelial and endothelial cell layers ("gut-on-a-chip)" treated with various concentrations of cytokines (TNF-a and IFN-g) to model intestinal inflammation in inflammatory bowel disease (IBD). Their outcomes included cytokine production (measured by ELSIAs) and intestinal barrier integrity (measured by permeability tracers). They demonstrate various cytokine concentrations and exposure conditions (top vs bottom channels) associated with specific cytokine production and that treatment of bottom channel (endothelial layer) with 100 ng/mL of proinflammatory cytokines decreased intestinal barrier integrity. The authors conclude that "induction of inflammation using only pro-inflammatory cytokines TNF-α and IFN-γ can create a simple and more accurate model of the cytokine secretion and barrier integrity aspects of IBD..." This is an interesting study that aims to provide a simplified, reductionist approach to study epithelial and endothelial dysfunction (cytokine production and barrier integrity) in inflammatory bowel disease (IBD). I have the following critiques and recommendations:

1. Abstract: "Our results indicate that this model consistently induces well-known IBD phenotypes." The authors should clarify what they mean by phenotype. Are they referring to the different IBD subtypes (ulcerative colitis vs Crohn's disease) or IBD phenotype behavior (stricturing vs inflammatory vs fistulizing/penetrating disease)? If they are referring to molecular phenotypes, they do not actually provide any transcriptomic/gene expression data to show that cytokine treatment of their epithelial/endothelial co-cultures recapitulates transcriptomics of epithelial and endothelial cells from biopsies from patients with IBD.

2. Introduction: "Despite the availability of multiple palliative drug options such as amino salicylates, glucocorticoid (GC), immunosuppressive (such as azathioprine methotrexate), and TNF-α monoclonal antibodies, a large proportion of patients either do not respond or lose response to therapy.." Would avoid the use of "palliative" as the IBD therapies mentioned above are effective treatments in attenuating intestinal inflammation in patients with IBD as opposed to providing only symptomatic relief/comfort.

3. Methods: The authors should clarify if their in vitro gut-on-a-chip model utilized the commercially available Emulate models. If so, this should be clearly stated in the methods section. Furthermore, they should report any financial disclosures and conflicts of interest and clarify role of the company in this manuscript.

4. Discussion: The authors should discuss the the strengths and limitations of their study. Cytokine treatment of epithelial and endothelial cells in vitro is not a novel concept. How does this particular study differ from prior studies (novelty)? Epithelial barrier dysfunction is only one aspect of IBD pathogenesis.

5. Discussion: How do the concentrations of TNF-a and IFN-g used in this in vitro model compare to in vivo tissue levels from animal models (e.g. DSS colitis) or human patients with IBD? Are these levels physiologic (and hence mimic IBD) and hence recapitulate intestinal inflammation in vivo?

6. Discussion: The authors should discuss the significance of their findings with regards to the different layer effects (epithelial layer-aspical/luminal vs endothelial-basolateral) with what is known from the literature. Presumably, cytokines from the bottom channel (in contact with endothelial layer) simulate cytokines derived from circulating/systematic immune cells, whereas cytokines from the top channel (in contact with epithelial cell layer) simulate apical/luminal-derived cytokines (from tissue resident immune cells). The spatial considerations of their model should be addressed with regards to what is known physiologically.

7. Results/Discussion: It is interesting that the authors are able to decrease intestinal barrier integrity. The authors should clarify in their results which barrier integrity (epithelial vs endothelial layer) was altered. Furthermore, the mechanism of decreased barrier integrity should be clarified. Was this mediated through loss of tight junction proteins? Was this mediated through apoptosis of epithelial and/or endothelial cells? The latter is an especially important point as IBD is characterized by erosions (resulting in death of epithelial cells) and ulcerations of the gastrointestinal tract.

Reviewer #2: In the present study Tataru et al. established and presented a gut-on-a-chip in vitro co-culture models enabling the study of factors and mechanisms of the non-immune inflammatory response of IBD. The data is clear and informative. However, there are some issues as described below.

Minor point

1. The authors described that currently existing experimental in vitro models can not examine factors that interrupt or decrease the “magnitude” of the non-immune inflammatory response characteristic of IBD in a part of “in vitro models of non-immune cell inflammatory response” in Introduction section. What is the magnitude? In the text that follows, the authors described that “we present a gut-on-a-chip co-culture model that stimulates non-immune cells with T-cell originating pro-inflammatory cytokines to simulate the non-immune inflammatory response.” The authors should describe not only the new experimental system, but also the magnitudes that could not be considered until now.

2. The authors cited paper number 18 in several places. However, the #18 paper is a review of experimental models of IBD, but not of an in vitro experimental system of IBD. Therefore, it is better to reconsider the cited papers for the places where the #18 paper is cited.

3. There are two nearly identical paragraphs in Conclusion section. There are slight differences, but do these differences mean anything? Authors need to clean up and rewrite the text of this section.

4. On page 3, line 76, “Ussing” is misspelled.

5. The markers of “Control” and “25 ng/ml top and bottom” are similar in color and difficult to distinguish in Fig. 2 and 3.

6. PLOS authors have the option to publish the peer review history of their article (what does this mean?). If published, this will include your full peer review and any attached files.

Reviewer #1: **Yes: **John Gubatan, MD

Reviewer #2: No

---

## [Author Response · Author response to Decision Letter 0]

5 Jul 2023

Reviewer's Responses to Questions

Comments to the Author

Reviewer #1: In this study, Tataru et al designed in vitro co-culture models of separated (by microporous membrane) epithelial and endothelial cell layers ("gut-on-a-chip)" treated with various concentrations of cytokines (TNF-a and IFN-g) to model intestinal inflammation in inflammatory bowel disease (IBD). Their outcomes included cytokine production (measured by ELSIAs) and intestinal barrier integrity (measured by permeability tracers). They demonstrate various cytokine concentrations and exposure conditions (top vs bottom channels) associated with specific cytokine production and that treatment of bottom channel (endothelial layer) with 100 ng/mL of proinflammatory cytokines decreased intestinal barrier integrity. The authors conclude that "induction of inflammation using only pro-inflammatory cytokines TNF-α and IFN-γ can create a simple and more accurate model of the cytokine secretion and barrier integrity aspects of IBD..." This is an interesting study that aims to provide a simplified, reductionist approach to study epithelial and endothelial dysfunction (cytokine production and barrier integrity) in inflammatory bowel disease (IBD). I have the following critiques and recommendations:

Abstract: "Our results indicate that this model consistently induces well-known IBD phenotypes." The authors should clarify what they mean by phenotype. Are they referring to the different IBD subtypes (ulcerative colitis vs Crohn's disease) or IBD phenotype behavior (stricturing vs inflammatory vs fistulizing/penetrating disease)? If they are referring to molecular phenotypes, they do not actually provide any transcriptomic/gene expression data to show that cytokine treatment of their epithelial/endothelial co-cultures recapitulates transcriptomics of epithelial and endothelial cells from biopsies from patients with IBD.

Thanks for this feedback. To clarify that we are only referring to cytokine profiles and barrier integrity, we’ve changed that sentence to read: “Our results indicate that this model 1. induces IBD-like cytokine secretion in non-immune cells and 2. decreases barrier integrity, making it a feasible and reliable model to test the direct actions of potential anti-inflammatory therapeutics on epithelial and endothelial cells.” 

Introduction: "Despite the availability of multiple palliative drug options such as amino salicylates, glucocorticoid (GC), immunosuppressive (such as azathioprine methotrexate), and TNF-α monoclonal antibodies, a large proportion of patients either do not respond or lose response to therapy.." Would avoid the use of "palliative" as the IBD therapies mentioned above are effective treatments in attenuating intestinal inflammation in patients with IBD as opposed to providing only symptomatic relief/comfort.

 Thanks, we have removed it!

Methods: The authors should clarify if their in vitro gut-on-a-chip model utilized the commercially available Emulate models. If so, this should be clearly stated in the methods section. Furthermore, they should report any financial disclosures and conflicts of interest and clarify role of the company in this manuscript.

The authors used the commercially available basic research kit from Emulate to develop their own organ-on-a-chip model and have no financial conflict of interest to disclose.. We have included the following sentence in the discussion section “Model usage” to clarify this issue:

“The described model was developed using the Emulate organ-on-a-chip basic research kit , (Emulate, MA).

Discussion: The authors should discuss the strengths and limitations of their study. Cytokine treatment of epithelial and endothelial cells in vitro is not a novel concept. How does this particular study differ from prior studies (novelty)? Epithelial barrier dysfunction is only one aspect of IBD pathogenesis.

Thanks for this comment, we’ve now expanded upon the advantages and limitations of the study in the discussion section “Advantages and limitations of the described system”

Discussion: How do the concentrations of TNF-a and IFN-g used in this in vitro model compare to in vivo tissue levels from animal models (e.g. DSS colitis) or human patients with IBD? Are these levels physiologic (and hence mimic IBD) and hence recapitulate intestinal inflammation in vivo?

We thank the reviewer for pointing out this issue. Establishing a correlation between in vitro and in vivo concentrations is highly challenging, especially taking into account differences in concentration between tissues (i.e. colon tissue vs. plasma) and differences between enzyme kinematics and bioavailability. In fact, when discussing drug metabolism it was found that the scaling factor of metabolic rate between an animal and in vitro cells could be as high as 1000 times (1).

To explore this question and see if we could compare concentrations appropriately, we conducted an extensive literature search. We found that in generality, mice treated with DSS or LPS to induce an IBD phenotype were found to have 0.1 - 2.1 ng/ml TNF-alpha in plasma, and 2 pg/mg - 750 pg/mg TNF-alpha in intestinal tissue (2-6). In contrast, Caco2 cells treated with TNF-alpha to induce an IBD-like response were treated with 10-100 ng/ml TNF-alpha (7-11). Thus, we tested the 0-100 ng/ml range of concentrations (provided in supplementary figure 4) and report responses relevant to IBD of various concentrations within that range.

Houston, J. Brian. "Utility of in vitro drug metabolism data in predicting in vivo metabolic clearance." Biochemical pharmacology 47.9 (1994): 1469-1479.

Berends, Sophie E., et al. "Tumor necrosis factor-mediated disposition of infliximab in ulcerative colitis patients." Journal of Pharmacokinetics and Pharmacodynamics 46.6 (2019): 543-551.

Xiao, Yong-Tao, et al. "Neutralization of IL-6 and TNF-α ameliorates intestinal permeability in DSS-induced colitis." Cytokine 83 (2016): 189-192.

Kriegel, Christina, and Mansoor M. Amiji. "Dual TNF-α/Cyclin D1 gene silencing with an oral polymeric microparticle system as a novel strategy for the treatment of inflammatory bowel disease." Clinical and translational gastroenterology 2.3 (2011): e2.

Murano, M., et al. "Therapeutic effect of intracolonically administered nuclear factor κ B (p65) antisense oligonucleotide on mouse dextran sulphate sodium (DSS)-induced colitis." Clinical & Experimental Immunology 120.1 (2000): 51-58.

Laroui, Hamed, et al. "Functional TNFα gene silencing mediated by polyethyleneimine/TNFα siRNA nanocomplexes in inflamed colon." Biomaterials 32.4 (2011): 1218-1228.

Apostolou, Athanasia, et al. "A Micro-engineered human Colon Intestine-Chip platform to study leaky barrier." Biorxiv (2020): 2020-08.

Barrenetxe, Jaione, et al. "TNFα regulates sugar transporters in the human intestinal epithelial cell line Caco-2." Cytokine 64.1 (2013): 181-187.

Le Phuong Nguyen, Thi, et al. "Protective effect of pure sour cherry anthocyanin extract on cytokine-induced inflammatory caco-2 monolayers." Nutrients 10.7 (2018): 861.

Contreras, Telma C., et al. "(−)-Epicatechin in the prevention of tumor necrosis alpha-induced loss of Caco-2 cell barrier integrity." Archives of biochemistry and biophysics 573 (2015): 84-91.

Juuti-Uusitalo, Kati, et al. "Differential effects of TNF (TNFSF2) and IFN-γ on intestinal epithelial cell morphogenesis and barrier function in three-dimensional culture." PLoS One 6.8 (2011): e22967.

 Discussion: The authors should discuss the significance of their findings with regards to the different layer effects (epithelial layer-apical/luminal vs endothelial-basolateral) with what is known from the literature. Presumably, cytokines from the bottom channel (in contact with endothelial layer) simulate cytokines derived from circulating/systematic immune cells, whereas cytokines from the top channel (in contact with epithelial cell layer) simulate apical/luminal-derived cytokines (from tissue resident immune cells). The spatial considerations of their model should be addressed with regards to what is known physiologically.

Thank you for the suggestion. We have added a section entitled “spatial considerations” to the discussion section exploring what is known in the literature about vectorial secretion and how it aligns with our own results.

Results/Discussion: It is interesting that the authors are able to decrease intestinal barrier integrity. The authors should clarify in their results which barrier integrity (epithelial vs endothelial layer) was altered. Furthermore, the mechanism of decreased barrier integrity should be clarified. Was this mediated through loss of tight junction proteins? Was this mediated through apoptosis of epithelial and/or endothelial cells? The latter is an especially important point as IBD is characterized by erosions (resulting in death of epithelial cells) and ulcerations of the gastrointestinal tract.

We thank the reviewer for their suggestion. Although the mechanism is yet to be elucidated, , we are confident that the observed permeability was not cell death-related. In traditional 2D culture, we performed a crystal violet cytotoxicity assay and determined that the cytokine concentrations used did not induce significant cytotoxicity in either cell line, and in fact seemed to increase growth in some concentrations. We’ve included this as a supplementary figure in the manuscript, thanks for the insightful question.

Reviewer #2: In the present study Tataru et al. established and presented a gut-on-a-chip in vitro co-culture models enabling the study of factors and mechanisms of the non-immune inflammatory response of IBD. The data is clear and informative. However, there are some issues as described below.

Minor point

The authors described that currently existing experimental in vitro models can not examine factors that interrupt or decrease the “magnitude” of the non-immune inflammatory response characteristic of IBD in a part of “in vitro models of non-immune cell inflammatory response” in Introduction section. What is the magnitude? In the text that follows, the authors described that “we present a gut-on-a-chip co-culture model that stimulates non-immune cells with T-cell originating pro-inflammatory cytokines to simulate the non-immune inflammatory response.” The authors should describe not only the new experimental system, but also the magnitudes that could not be considered until now.

Thank you for pointing out that the phrasing was unclear. We replaced “decrease the magnitude” by “reduce”;, hopefully this clarifies the sentence.

We have also modified the language of this paragraph to emphasize that while most models mainly focus on studying the barrier integrity, this system also enables daily cytokine secretion measurements.

The authors cited paper number 18 in several places. However, the #18 paper is a review of experimental models of IBD, but not of an in vitro experimental system of IBD. Therefore, it is better to reconsider the cited papers for the places where the #18 paper is cited.

Thanks for highlighting this issue. One of the citations was indeed incorrectly placed, and we have replaced it by reference #17 in this revised version of the manuscript. However, paper #18 described the DSS and LPS phenotypes very well, which is relevant to the context of the paragraphs where that citation was included. Paper #17 describes that similar treatments can be used in vivo and in vitro, while paper #18 describes the treatment effect on phenotype. We have included additional citations to this section that demonstrate the use of these DSS and LPS treatments in vitro, see below for convenience:

Gao, R., Shu, W., Shen, Y., Sun, Q., Bai, F., Wang, J., ... & Yuan, L. (2020). Sturgeon protein-derived peptides exert anti-inflammatory effects in LPS-stimulated RAW264. 7 macrophages via the MAPK pathway. Journal of Functional Foods, 72, 104044.

Aly, E., López-Nicolás, R., Darwish, A. A., Ros-Berruezo, G., & Frontela-Saseta, C. (2019). In vitro effectiveness of recombinant human lactoferrin and its hydrolysate in alleviating LPS-induced inflammatory response. Food Research International, 118, 101-107.

Maria-Ferreira, D., Nascimento, A. M., Cipriani, T. R., Santana-Filho, A. P., Watanabe, P. D. S., Luciano, F. B., ... & Baggio, C. H. (2018). Rhamnogalacturonan, a chemically-defined polysaccharide, improves intestinal barrier function in DSS-induced colitis in mice and human Caco-2 cells. Scientific reports, 8(1), 1-14.

There are two nearly identical paragraphs in Conclusion section. There are slight differences, but do these differences mean anything? Authors need to clean up and rewrite the text of this section.

We apologize for this mistake. We have fixed this issue.

On page 3, line 76, “Ussing” is misspelled.

Corrected.

The markers of “Control” and “25 ng/ml top and bottom” are similar in color and difficult to distinguish in Fig. 2 and 3.

Thank you for the comment, we changed the color to a light pink which should increase visibility.

---

## [Decision Letter · Decision Letter 1]

17 Jul 2023

Cytokine induced inflammatory bowel disease model using organ-on-a-chip technology

PONE-D-23-02456R1

Dear Dr. Christine Andrea Tataru,

We’re pleased to inform you that your manuscript has been judged scientifically suitable for publication and will be formally accepted for publication once it meets all outstanding technical requirements.

Kind regards,

Masanori A. Murayama

Academic Editor

PLOS ONE

Additional Editor Comments (optional):

Thank you for submitting revised manuscript. In this time, I will decide your paper "Accept". Congratulations.

Reviewers' comments:

Reviewer's Responses to Questions

**Comments to the Author**

1. If the authors have adequately addressed your comments raised in a previous round of review and you feel that this manuscript is now acceptable for publication, you may indicate that here to bypass the “Comments to the Author” section, enter your conflict of interest statement in the “Confidential to Editor” section, and submit your "Accept" recommendation.

Reviewer #1: All comments have been addressed

Reviewer #2: All comments have been addressed

2. Is the manuscript technically sound, and do the data support the conclusions?

Reviewer #1: Yes

Reviewer #2: Yes

3. Has the statistical analysis been performed appropriately and rigorously? 

Reviewer #1: Yes

Reviewer #2: Yes

4. Have the authors made all data underlying the findings in their manuscript fully available?

Reviewer #1: Yes

Reviewer #2: Yes

5. Is the manuscript presented in an intelligible fashion and written in standard English?

Reviewer #1: Yes

Reviewer #2: Yes

6. Review Comments to the Author

Reviewer #1: The authors have adequately addressed my critiques. I appreciate the insightful responses and additional experiments performed.

Reviewer #2: The authors carefully responded to comments and questions from reviewers. As a result, the content of this paper is acceptable.

7. PLOS authors have the option to publish the peer review history of their article (what does this mean?). If published, this will include your full peer review and any attached files.

Reviewer #1: **Yes: **John Gubatan, MD

Reviewer #2: No

---

## [Editor Report · Acceptance letter]

20 Jul 2023

PONE-D-23-02456R1 

Cytokine induced inflammatory bowel disease model using organ-on-a-chip technology 

Dear Dr. Tataru:

I'm pleased to inform you that your manuscript has been deemed suitable for publication in PLOS ONE. Congratulations! Your manuscript is now with our production department. 

Kind regards, 

on behalf of

Dr. Masanori A. Murayama 

Academic Editor

PLOS ONE